# Colorimetric Quantification for Residual Poly-DADMAC in Water Treatment

**Ilil Levakov [1], Ido Maor [1,2,†], Chen Barak [1], Yael Kirshenbaum [2] and Giora Rytwo [1,2,\***

[1] Environmental Physical Chemistry Laboratory, MIGAL-Galilee Research Institute, Kiryat Shmona 1101602, Israel; ilill@migal.org.il (I.L.)
[2] Environmental Sciences & Water Sciences Department, Tel Hai College, Upper Galilee 1220800, Israel
\* Correspondence: rytwo@telhai.ac.il or giorarytwo@gmail.com; Tel.: +972-4-7700516
† Deceased.

**Abstract:** Poly-DADMAC (PD) is a commonly used organic polymer in water treatment, known for its effectiveness as a coagulant. However, its presence as a residue in water raises concerns related to membrane fouling and the potential formation of carcinogenic compounds. Therefore, fast and simple quantification is necessary to efficiently control and monitor the optimal dose of poly-DADMAC with minimal negative effects. This study introduces a new colorimetric quantification method for poly-DADMAC, based on complexation with a cationic dye (fast green-FG). The method was examined through varying conditions, which included different analytical and commercial poly-DADMAC formulations and concentrations. These experiments confirm its effectiveness in quantifying poly-DADMAC with a detection limit of 3.22 $\mu g\ L^{-1}$ (0.02 $\mu M$ based on monomers' molecular weight), which is one order of magnitude lower than regulatory requirements (50 $\mu g\ L^{-1}$). To validate the method, the effect of pH was examined, and implementation demonstrations were conducted on cyanobacteria and cowshed-polluted water samples. This research introduces a fast, cost-effective innovative method to accurately quantify poly-DADMAC, enhancing water treatment strategies for high-quality purification and water reuse

**Keywords:** cationic polyelectrolyte coagulant; fast green; spectrophotometric quantification; water treatment





## 1. Introduction

Coagulation and flocculation are important preliminary steps in wastewater and potable water treatment facilities. The processes include the removal of non-dissolved and particulate matter, including algae, pathogens, organic materials, minerals, and others. This is achieved by the addition of materials, such as ferric chloride, alum (aluminium sulfate), or organic polymers, which destabilize the colloids resulting in small suspended particles agglomerating into larger settleable flocs [1]. Poly-diallyl-dimethylammonium chloride (poly-DADMAC, PD, CAS 26062-79-3) is a homopolymer also known as polyquarternium-6, with a chemical formula of $(C_8H_{16}NCl)_n$ (Figure 1). In commercial formulations, its molecular weight typically ranges from a few hundred to several thousand kDa, depending on the number of monomers (each with a molecular weight of 161.7 Da) [2]. PD is the most common polymer used in various coagulation processes [3], including in the pulp and paper industry, either alone [4] or in combination with other polymers [5]. It was the first cationic polyelectrolyte approved by the Food and Drug Administration of the U.S.A. (FDA) for use in potable water treatment [6]. Recent studies have also tested its influence when adsorbed on cellulose-based products [7] or for the preparation of specific adsorbing matrices for per- and polyfluoroalkyl substances [8].

**Figure 1.** Chemical structure of polyDADMAC (PD, polyquaternium-6, **left**) and fast green (FG, E143, **right**).

The wide utilization of PD and other organic polymers can be attributed to several advantages. These include their remarkable cationic properties, high-charge density that leads to the formation of stronger and easily separable flocs, effective bridging capabilities allowing for higher solid content in the sludge phase, and minimal impact on pH levels, necessitating only a single pH adjustment during the process. However, a few disadvantages are associated with the use of PD as a coagulant, mainly regarding the residuals remaining after the process. While PD can serve as an effective coagulant, its presence in the subsequent treatment process might cause severe fouling to the filtration membranes. Although novel membranes with better anti-fouling and anti-microbial properties have been developed [9,10], residual PD molecules of even low concentrations can desorb onto the membrane surface, potentially leading to pore blocking [11]. Fouling depends on the type, charge, size, and concentration of the polymer, and higher molecule-weight polymers have larger fouling potential [12,13]. Membrane fouling, particularly in processes like reverse osmosis, microfiltration, and nanofiltration, can lead to reduced water flux and increased energy consumption. It necessitates more frequent membrane cleaning or replacement, subsequently elevating operational costs in water treatment plants [14]. An additional disadvantage is the reaction of residual PD with disinfection by-products, such as chloramine and ozonation, forming carcinogenic nitrosamines [15–19]. Therefore, several American and European standardization organizations have limited the residual amount of PD in drinking water at $\leq 50$ µg/L~0.310 µM, which is considered to be also toxic to aquatic organisms [20].

For these reasons, fast and simple quantification of residual PD is required at relatively low limits of detection (LOD), to control and monitor the optimal coagulant dose with minimal negative effects. Chromatography-based analytical methods for the quantification of polymers are usually very sensitive to interferences by other organic molecules [21]. Indeed, a recent study presented a method based on ion-pair chromatography, which accurately measured residual DADMAC monomers [22] without measuring the amounts of the polymer. Other accurate methods, such as gel permeation chromatography, epoxidation, fluorescent tagging, and co-precipitation, require advanced, relatively expensive, and long procedures [2,3,23–26]. A highly sensitive method was developed based on the use of gold nanoparticles [27], applied in South African treated water [28] and recently developed to Lovibond color filters [29], but the preparation of the nanoparticles requires a relatively complicated procedure.

Cationic polyelectrolytes, such as PDs, can bind to anionic molecules, such as tannic acid or anionic dyes like rose Bengal, methyl-orange [23], Ponceau S [30], and acid orange [31], resulting in an insoluble polymer-anion complex that can be separated by centrifugation. This separation method allows the quantification of the polymer concentration by measuring the remaining dye. A similar method was used for quantification of the cationic biopolymer chitosan, based on complexation with Cibacron Brilliant Red 3B-A [32]. However, in all those studies, LODs were two to three orders of magnitude higher than the requirement.

In this study, we present a simple colorimetric quantification method based on the addition of fast green (Figure 1, FG, ethyl-[4-[[4-[ethyl-[(3-sulfophenyl) methyl] aminophenyl]-(4-hydroxy-2-sulfophenyl)methylidene]-1-cyclohexa-2,5-dienylidene]-[(3-sulfophenyl)methyl] azanium), and its complexation with PD. FG is an anionic tri-aryl methane food dye known also as "food green 3", "green 1724", and "E143". It is stable in a wide range of pH and has a brilliant blue-green color with an absorption maximum of 624 nm, and two additional smaller absorption bands at 420 and 304 nm [33]. The very high molar absorptivity ($\varepsilon_{624} > 100,000$ $M^{-1}cm^{-1}$) enables high sensitivity and yields a procedure suitable for trace quantification of PD with LOD < 3 $\mu$g $L^{-1}$ (0.018 $\mu$M), one order of magnitude lower than regulation requirements. The primary objective of this research was to establish the proof of concept for the proposed quantification method in a simple, fast, and cost-effective manner. This approach serves as the first step in mitigating PD's impact on membrane fouling and potential environmental risks. Additionally, we conducted preliminary implementations of the method under different conditions.

## 2. Materials and Methods

### 2.1. Materials

Analytical polydiallyl dimethylammonium chloride (PD; medium molecular weight, 200,000 to 350,000) and fast green (FG) were purchased from Sigma-Aldrich (Jerusalem, Israel). A commercial-grade coagulant with 40% PD as an active ingredient manufactured by SNF Ltd. (FLOQUAT® FL-45) was purchased from Amgal-Depotchem Ltd. (Beer Tuvia, Israel), SNF representatives in Israel. NC24 clay polymer nanocomposites were prepared as described in the literature [34–36] using 10 g/L sepiolite S9 provided by Tolsa S.A. (Madrid, Spain), and 45 g/L FL-45. DKG kaolinite clay polymer nanocomposites [37] were prepared in a similar way, using 10 g/L MPO kaolinite provided by Agat Minerals and Yehu Clays Ltd. (Yeruham, Israel) and 45 g/L FL-45. A total of 1 mM HCl was prepared from 37% stock (Merck, Rehovot, Israel) and used for pH adjustment.

### 2.2. Complexation/Calibration Experiments Description

In order to prepare calibration curves and demonstrate the method's applicability, several complexation experiments were conducted, including a wide range of analytic poly-DADMAC with fast green dye. Three separate experiments were conducted to examine different analytic PD concentrations. These concentrations included low, medium, and high PD ranges: 0.02–1 $\mu$M (0.0322–0.1617 mg $L^{-1}$), 1–30 $\mu$M (0.1617–4.851 mg $L^{-1}$), and 10–200 $\mu$M (1.617–32.34 mg $L^{-1}$) PD solutions, all of which were dissolved in double-distilled water. To each PD range, 0.8, 20, and 100 $\mu$M of FG were added to the solutions in the low, medium, and high series, respectively. After the addition of FG, pH values were adjusted to 3–3.5 by adding 20 $\mu$L of 1 mM HCl (see Section 2.4). Subsequently, all samples were kept at room temperature (23 ± 1 °C) on an orbital shaker (200 rpm) for 1 h. FG spectra were measured using a 1 cm cuvette in a Cary 60 UV–Vis spectrophotometer (Agilent Technologies, Santa Clara, CA, USA). Absorbance was determined using the absorption bands at 624 and 420 nm with a molar absorptivity of 103,500 and 11,600 $M^{-1}cm^{-1}$, respectively. To enhance sensitivity, absorbances of the low-range FG solutions were measured at 624 nm, utilizing a 5 cm cuvette. A visual representation of the preparation procedure and the experimental setup is available in Figure S1 of the Supplementary Material.

To confirm the formation of the PD-FG complexes, 50 mL samples containing 20 $\mu$M FG and a range between 2.5 and 50 $\mu$M PD solutions were prepared as described above, centrifuged, and the sediment was lyophilized (Christ Alpha 1-2 LD Plus, Germany) and measured for the attenuated total reflection-Fourier transform infrared (ATR-FTIR) spectra on a Nicolet iS10 FTIR (Nicolet Analytical Instruments, Madison, WI, USA), using a SMART ATR device with a diamond crystal plate, within a range of wavenumber comprised from 4000 to 525 $cm^{-1}$ and analyzed with OMNIC 9.3.32 software (Thermo Fisher Scientific, Madison, WI, USA), including full spectra ATR correction.

*2.3. Implementation in Water with Commercial PD*

Commercial PD samples were examined in medium-range concentrations of 2–50 μM (0.323–8.085 mg L$^{-1}$) PD dissolved in double-distilled water with the addition of 20 μM FG. FG spectra were measured in the UV–Vis spectrophotometer using a 1 cm cuvette as described in Section 2.2.

*2.4. Influence of pH*

To assess the performance of the quantification method in actual water samples, two coagulation-flocculation experiments were conducted. These experiments were carried out using water contaminated with cyanobacteria and cowshed effluents as test samples. The evaluation involved testing both a commercial PD formulation and clay-PD nanocomposites developed within our research group.

*2.5. Quantification of Poly-DADMAC in Coagulation Water Treatment Samples*

To evaluate the efficiency of the quantification method in "real" water samples, two coago-flocculation experiments were performed in cyanobacteria-polluted water and cowshed effluents. Experiments included testing commercial PD formulation, and clay-PD nanocomposites developed in our research group [35–37]

The first coagulation experiment was performed on cyanobacteria culture suspension using concentrations of two poly-DADMAC-based coagulants: industrial PD and kaolinite-PD nanocomposites (DKG). The cyanobacteria suspension consisted of the Microcystis family, which was isolated from Lake Kinneret and cultivated in batch cultures using BG11 medium [38]. The experiments were carried out in 300 mL glass beakers and included two sets of PD and DKG coagulants. Each set comprised five distinct coagulant concentrations: 14, 28, 42, 53, and 71 μM PD, either in its free form or pre-adsorbed onto kaolinite clay particles (referred to as DKG coagulant). The coagulants were added to 250 mL cyanobacteria suspensions. After the addition, the suspensions were stirred thoroughly for 1 min and left to settle for 1 h.

The second coagulation experiment was performed on cowshed effluents. The wastewater was taken from the Kfar Blum industrial cowshed (Upper Galilee, Israel) in July 2023. To determine the required amount of coagulant needed for neutralization [39], the electrokinetic colloidal charge of the wastewater was measured in a particle charge detector (PCD) (BTG Mütek, PCD-05, Eclépens, Switzerland). Three poly-DADMAC-based coagulants were tested: commercial PD, kaolinite-PD nanocomposite (DKG), and sepiolite-PD nanocomposite (NC). The doses were equivalent to 100% of the requirement to neutralize the colloids (4.3 mM charges). Each experiment was conducted in 300 mL glass beakers, with each treatment having 3 replicates. After the addition, the suspensions were thoroughly stirred for 1 min and allowed to settle for 1 h

In both experiments, turbidity was measured with a LaMotte 2020i turbidimeter (Chesterton, MD, USA) before and after the treatment to estimate the efficiency of clarification and coagulation. The residual PD concentrations were quantified using the complexation method as described in Section 2.2. Specifically, 10 mL of the supernatant was sampled, and the pH values were adjusted to 3–3.5 by adding 20 μL of 1 mM HCl. Subsequently, suitable volumes of FG from a 1000 μM stock solution were added to each sample, yielding an FG concentration of 20 μM. Test tubes were placed on an orbital shaker (200 rpm) for 1 h, and samples were measured in the spectrophotometer as described in Section 2.2.

## 3. Results

*3.1. Quantification of Analytical Poly-DADMAC Concentrations*

As mentioned in Section 2.2, three separate calibration experiments were conducted to quantify PD at different concentration levels: low, medium, and high. Each range included the addition of different FG concentrations. The quantification method is based on the measurement of free fast-green molecules in the solution, according to its distinct absorption bands at wavelengths 420 and 624 nm. Figure 2 shows the measured absorbance

values in specific wavelengths as a function of the known PD concentrations. To increase sensitivity at the medium and low PD range, the selected wavelength was 624 nm. To avoid the need for dilutions at the high range the 420 nm (with lower molar absorptivity) was selected. Linear regression calculations between absorbance and PD concentrations were performed for all three ranges (see Figure 2), showing excellent correlations ($R^2 > 0.982$) in the high and medium ranges (1–200 μM, 0.1617–32.4 mg $L^{-1}$). According to the results, as the PD concentration increases, absorbance values decrease, indicating the formation of a PD-fast green complex that reduces available fast green molecules in the solution. This inverse relationship between dye absorbance and PD concentration serves as the foundation for quantitatively analyzing PD through complexation with the FG dye, similar to the effect observed in the chitosan quantification method [32]. The relatively low $R^2$ ($R^2 = 0.866$) in the 0.02–1 μM PD range is attributed to increased measurement noise and experimental variability at these low concentrations. The lower PD concentration used for the calibration experiment was determined as the detection limit of the method (0.02 μM).

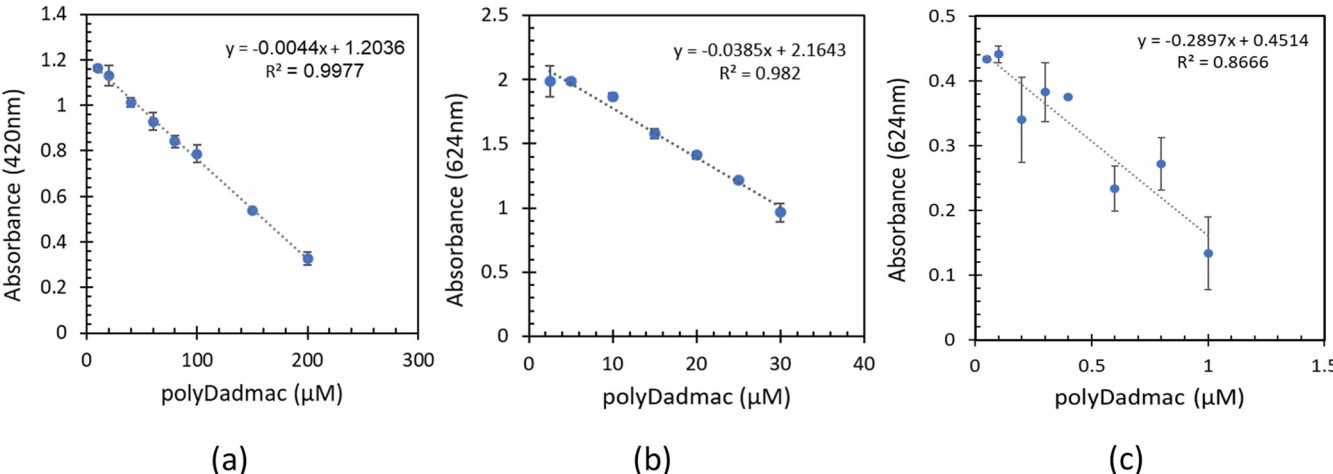

**Figure 2.** Absorbance values as a function of poly-DADMAC concentrations after complexation with fast green. (**a**) High PD range, $OD_{420}$ nm of 10–200 μM PD upon addition of 100 μM FG. (**b**) Medium PD range, $OD_{624}$ of 1–30 μM PD upon addition of 20 μM FG. (**c**) Low PD range, $OD_{624}$ of 0.02–1 μM PD upon addition of 0.8 μM FG. High and medium ranges were measured in a 1 cm cuvette, whereas low range—in a 5 cm cuvette.

As a confirmation step, the necessity of removing the PD-fast green complex from the solution by centrifugation was tested. The samples underwent centrifugation at 1100× *g* for 20 min, followed by spectrophotometer measurements. According to the results, centrifugation of the mixture solution after complexation did not affect the absorbance values (data not shown).

### 3.2. FTIR Measurements of PD-FG Complexes

Figure 3 shows the FTIR spectra of dried PD, FG, and sediments of samples with 20 μM FG complexed with 2.5 and 50 μM PD. The raw PD spectrum shows a distinct absorption band at 1464 $cm^{-1}$, attributed to quaternary ammonium bound to chloride anions [40]. Additionally, a broad peak associated with aliphatic secondary amines at 3100–3600 $cm^{-1}$ [26] and bands related to C-H stretching vibrations of methine, methylene, and methyl groups [41] at 3000–2800 $cm^{-1}$ can be clearly observed. The raw FG dye spectrum exhibits a series of absorption bands related to aromatic structures and sulfonate vibrations [41] at 1340 and 1170 $cm^{-1}$ In both the FG-2.5 μM PD and FG-50 μM PD complexes, the vibrations associated with quaternary ammonium bound to chloride anions completely disappear, suggesting that the complexation occurs through the exchange of chloride ions with the anionic dye. The spectra also display distinct bands corresponding

to the C-H and secondary amine groups of PD, along with aromatic and sulfonate bands from FG. These observations confirm the formation of the dye-coagulant complex. Similar spectra were recorded for all measured FG-PD complexes, and shown in Figure S2 in the Supplementary Material.

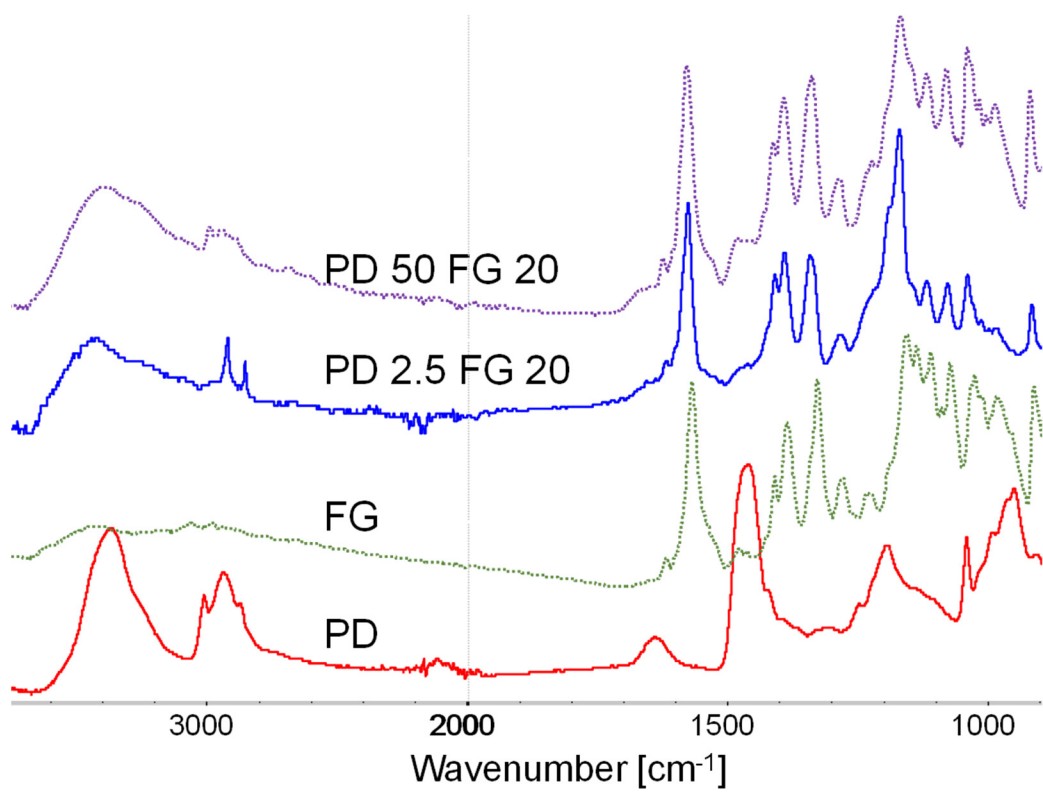

**Figure 3.** FTIR spectra of dried polyDADMAC (PD, red line), raw fast green (FG, green line), sedimented complexes formed by 2.5 (PD 2.5 FG 20, blue line), and 50 μM PD (PD 50 FG 20, purple line). Note that the x-axis is split at 2000 cm$^{-1}$.

*3.3. Influence of pH*

Given that the accuracy of spectroscopy measurements relies on the stability of the spectrum, we conducted experiments to assess the impact of pH on FG spectra. The absorbance spectrum of FG was measured at different pH values ranging from 2 to 9.1 (Figure 4). The results indicate that within the pH range of 2 to 7, the spectrum of FG remained constant, with no changes in the molar absorptivity at any wavelength. This suggests that there were no significant alterations in the chromophore within this acidic to neutral pH range. However, as the pH increased above 7, the spectrum deviated from the constant trend observed in the acidic to neutral pH range. The large absorption band at 624 nm shifted hypsochromically (to lower wavelengths), and molar absorptivity slightly increased, while the absorption band at 420 nm disappeared. These findings are crucial to determine the pH requirements and limitations relevant when applying the quantification method presented hereby and highlight the importance of pH control to a range of 2–7 during the analysis process. In our study, the pH of all samples was adjusted to a range of 3–3.5 by adding very small volumes of concentrated HCl before measuring the spectrum

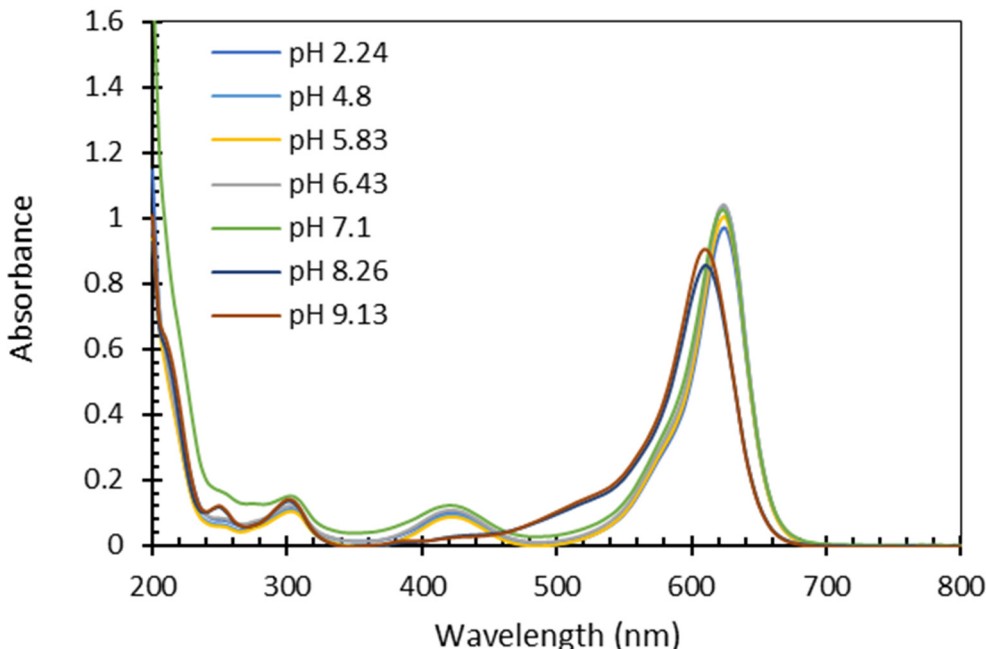

**Figure 4.** UV-Vis spectra of 10 µM FG solutions at different pH values (2.24–9.13).

*3.4. Quantification of Commercial Poly-DADMAC*

Complexation experiments were conducted with commercial PD in a range of 1–50 µM polymer concentrations, with the addition of 20 µM FG according to the medium range presented in Section 3.1. Figure 5 shows the absorbance of the remaining dye as a function of commercial PD concentrations, after the formation of the complexes at 420 (Figure 5a) and 624 nm (Figure 5b). To prevent any potential shift in the spectrum due to high pH, all samples were acidified with 20 µL of 1 mM HCl, as outlined in Section 3.3. The full spectra of commercial PD samples after complexation with FG are shown in Figure S3 in the Supplementary Material. To verify the similarity of the measured spectra to that of FG, a Pearson's correlation of all spectra with FG in the 300–800 range nm was evaluated using the CORREL function in Excel©, yielding R > 0.993.

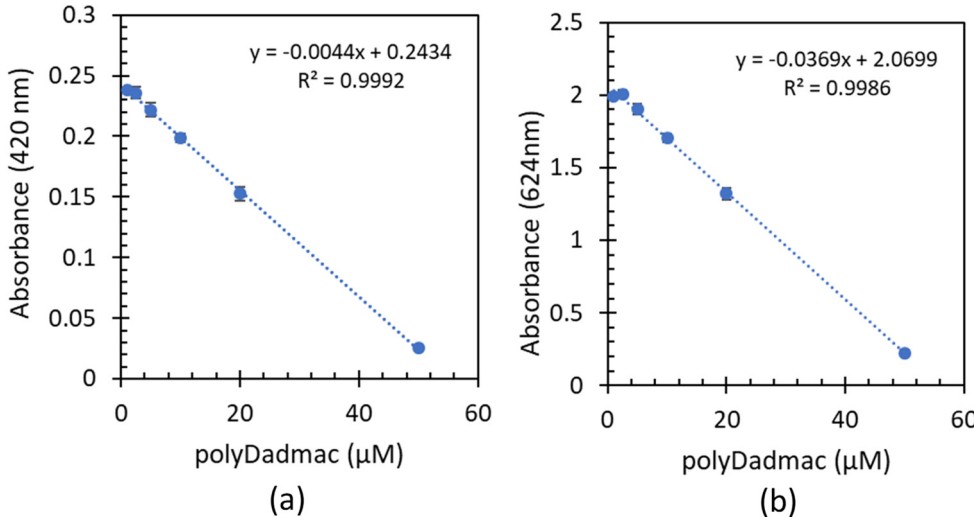

**Figure 5.** Absorbance values at (**a**) 420 nm and (**b**) 624 nm, as a function of PD concentrations (1–50 µM) after complexation with 20 µM FG.

Consistent linear relationships between absorbance and PD concentration were observed at both wavelengths (R2 = 0.9992 for 420 nm and R2 = 0.9986 for 624 nm), indicating a strong and reliable correlation, as illustrated in Figure 2 for analytical PD. It is noteworthy that after the examination of the linear regression equations in Figures 2b and 4b, a pronounced similarity emerges, both in terms of the intercept and slope values. Furthermore, the slope in Figures 2a and 5a are identical, although the intercept differs due to levels of FG concentrations. These findings highlight the applicability of the complexation method for quantifying commercial PD concentrations.

*3.5. Quantification of Poly-DADMAC in Coagulation Treatment Samples*

The method's applicability in practical field conditions was assessed by determining residual PD concentrations in cyanobacteria-contaminated water and cowshed effluents during coagulation and clarification treatments.

The cyanobacteria water samples were treated with commercial PD and kaolinite-PD nanocomposites (DKG). As mentioned in Section 2.5, coagulants were added in amounts equivalent to the following PD concentrations: 14, 28, 42, 53, and 71 µM (either in free form or adsorbed onto kaolinite clay particles). The initial turbidity value for the cyanobacteria suspension was 98.2 ± 3.67 NTU. Following the coagulant addition as detailed in Section 2.5, 10 mL test tubes were used to sample the supernatants. These samples were then acidified, and a concentrated FG solution was added, resulting in an FG concentration of 20 µM. The absorbance was measured at 420 nm, and the residual PD concentration was determined using the calibration curve established in the previous section (Figure 5). Residual PD concentrations and turbidity values are shown in Figure 6. Based on the measured results, minimal turbidity is observed across the entire range of added PD-kaolinite, and up to 53 µM of commercial PD. Additionally, there is a substantial clear phase, accounting for approximately 90% of the volume, at the top of the vessels. However, increasing residual PD is clearly observed as the added amounts of coagulant increase. Furthermore, significant differences between the remaining PD concentrations were observed between the two coagulants, with the lowest amounts of 1.9 and 4.1 PD µM for the nanocomposites and commercial PD, respectively. The lower residual PD values observed in the DKG samples are likely attributed to the PD binding to the kaolinite clay particles and the higher density of these clays. This, in turn, leads to the enhanced precipitation of excess coagulant materials. As the coagulant concentrations increased, no significant changes were observed in the turbidity values. However, higher turbidity was observed in the addition of 71 µM commercial PD indicating coagulant over-dose that disrupts the neutralization of the colloidal charge of the suspension [39].

As an additional test for the method, we conducted clarification and coagulation experiments on cowshed effluents. Such effluents are heavily polluted with a total suspended solid (TSS) value that may reach 10 g L$^{-1}$, and turbidity >4000 NTU [42]. TSS removal is required as pretreatment since wastewater treatment plants limit the maximum inflow value to <0.4 g L$^{-1}$ [43]. In the samples collected at the Kfar Blum cowshed during July 2023, the initial turbidity of the effluents was 3355 ± 226 NTU. Optimal coagulant doses were determined by measuring the colloidal particle charge as described in Section 2.5. Following the coagulation process, turbidity decreased to 1.2–4.5% of the initial values (77, 152, and 43 NTU in the kaolinite-PD, sepiolite-PD, and commercial PD coagulant treatments, respectively). The figure below presents the residual PD concentrations in the three treatments (Figure 7). As in the cyanobacteria-polluted effluents, residual PD was considerably lower in the clay-PD coagulants (~25 µM = ~4.04 mg L$^{-1}$) than in the commercial PD (37 µM = 5.98 mg L$^{-1}$), probably due to the high density of the clays and precipitation of excess coagulants.

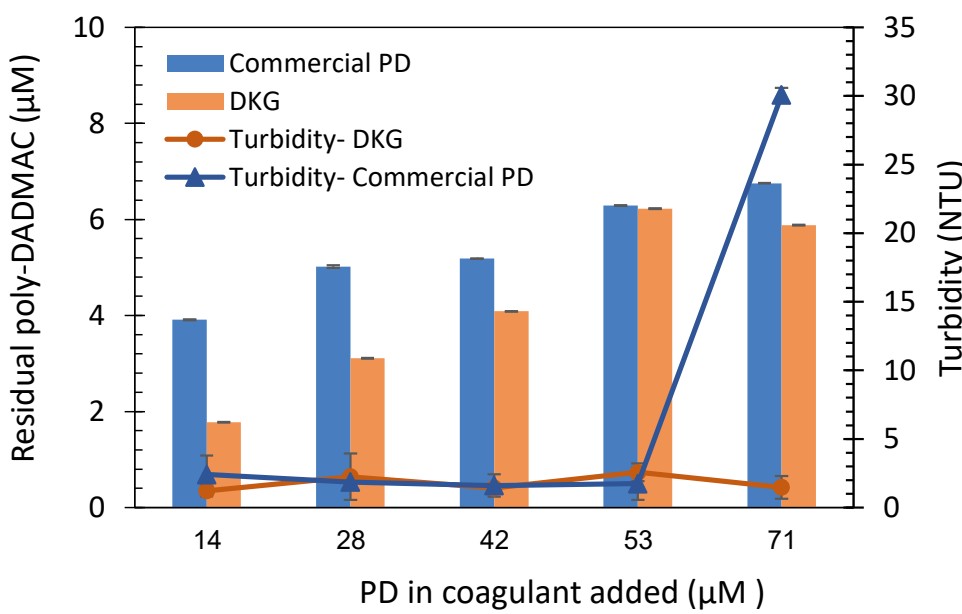

**Figure 6.** Evaluated PD residual concentrations (bars) and measured turbidity (lines and dots) in cyanobacteria-polluted water at different added PD doses, using commercial PD (blue, triangles) or kaolinite-PD nanocomposite (orange, circles).

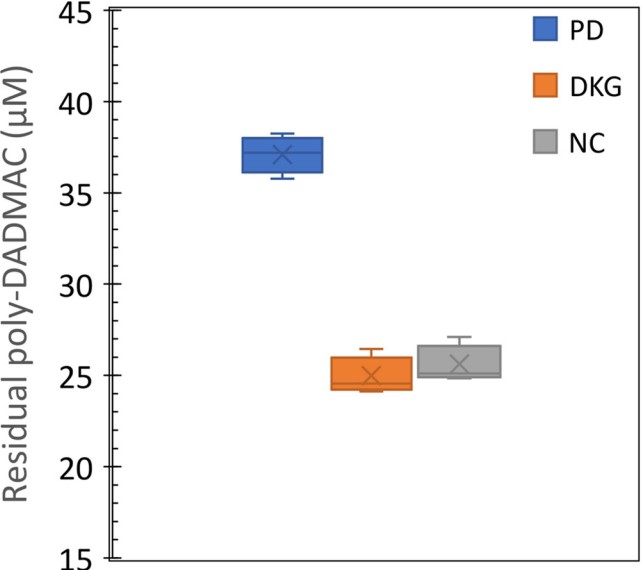

**Figure 7.** Residual PD in cowshed effluent coagulation experiments with commercial PD (blue), kaolinite-PD nanocomposites-DKG (orange), and sepiolite-PD nanocomposites-NC (grey).

Normalized spectra of cyanobacteria and cowshed water samples after treatment compared to the spectrum of FG with commercial PD in clear water are presented in Figure S4 of the Supplementary Material. FG spectrum in cyanobacteria sample is almost identical to clear water (Pearson's correlation in the range 300–800 nm = 0.999). On the other hand, FG in the cowshed effluents differs from that in clear water (Pearson's correlation = 0.938) due to the presence of remaining color and turbidity. However, limiting the range to 600–800 nm, where colloids light dispersion is lower, increases Pearson's correlation to 0.977, indicating that differences do not interfere with the measurement of absorbance at 624 nm.

The results of both demonstration experiments highlight the method's applicability in real conditions, offering vital insights into optimal coagulation treatments in a convenient,

cost-effective manner. It should be mentioned that the removal of residual PD concentration from post-coagulation wastewater is essential for the prevention of extensive membrane fouling and negative environmental impact. Several studies examined different approaches in order to reduce the negative impact of PD by inhibiting the formation of carcinogenic by-products [19,44], but additional studies are required to examine possible complete removal techniques, such as adsorption and ion exchange. Nevertheless, the first step to approach the problem is to improve the monitoring ability of PD in wastewater plant facilities.

## 4. Conclusions

- In this study, a colorimetric quantification method for poly-DADMAC through complexation with fast green dye was developed and evaluated.
- The method exhibited high sensitivity with a detection limit of 0.02 μM (0.0032 mg L$^{-1}$), meeting regulatory demands according to the world standard limitations.
- Quantification experiments of analytical PD concentrations, including low, medium, and high ranges demonstrated a linear correlation between the absorbance and PD concentrations.
- The influence of pH on the quantification process was tested, revealing that the absorbance spectrum of FG remains unchanged within the pH range of 2 to 7, exhibiting a hypsochromic and hyperchromic shift at higher pH. Therefore, an acidification step (or accurate pH measurements) is essential before quantification to ensure the FG spectrum falls within the required range.
- The quantification method's feasibility was demonstrated in coagulation experiments, estimating remaining PD concentrations in treated water samples.
- The method's feasibility in estimating PD concentrations under actual coagulation conditions highlights its potential for efficient monitoring and control of coagulation processes. It offers a rapid, cost-effective, and sensitive tool for accurate PD concentration measurement. The colorimetric PD quantification holds promising potential for advancing water treatment systems and quality control.
- Further research may concentrate on upscaling the application of the quantification method for large-scale water treatment systems, as well as refining its adaptation and addressing limitations. This may include examining factors such as temperature, reaction time, and ionic strength.

**Supplementary Materials:** The following supporting information can be downloaded at: https://www.mdpi.com/article/10.3390/w15193352/s1, Figure S1: Scheme of the experimental setup. Figure S2: ATR-FTIR spectra of PD-FG complexes. Figure S3: Spectra of an initial 20 μM FG upon complexation with the increasing concentrations of commercial PD; Figure S34: Normalized spectra of an initial 20 μM FG upon complexation with commercial PD in distilled water, cyanobacteria polluted water and cowshed effluents.

**Author Contributions:** Conceptualization, G.R.; methodology, G.R., I.L., C.B. and I.M.; software, I.L. and Y.K.; validation, G.R., I.L. and Y.K.; formal analysis, I.L.; investigation, G.R., I.L., C.B. and I.M.; resources, G.R.; data curation, I.L. and G.R.; writing—original draft preparation, G.R., I.L. and C.B.; writing—review and editing, I.L. and G.R.; visualization, I.L. and Y.K.; supervision, G.R.; project administration, G.R.; funding acquisition, G.R. All authors have read and agreed to the published version of the manuscript.

**Funding:** This research was partially funded by CSO-MOH (Israeli Ministry of Health), in the frame of the collaborative international consortium (REWA) financed under the 2020 AquaticPollutants Joint call of the AquaticPollutants ERA-NET Cofund (GA No. 869178).

**Data Availability Statement:** All raw data are available from the authors.

**Acknowledgments:** We would like to dedicate this manuscript to the memory of the late Ido Maor. The authors would like to thank the European Commission and AKA (Finland), CSO-MOH (Israel), IFD (Denmark), and WRC (South Africa) for funding in the frame REWA international consortium (additional details in "funding" paragraph). REWA is an integral part of the activities developed by

the Water, Oceans and AMR JPIs. The authors are also thankful to the Yehezkel Zwecher for growing the cyanobacteria suspension, Simmah Hendrickson for the English editing, and the whole team of the Hydrogeology and Examination of Soil Fertility Lab at MIGAL Research Institute.

**Conflicts of Interest:** The authors declare no conflict of interest.

## Abbreviations

| | |
|---|---|
| PD | poly-dadmac (Poly-diallyl-dimethylammonium chloride) |
| FG | Fast-green |
| LOD | limit of detection |
| DKG | kaolinite-PD nanocomposites |
| NC | sepiolite-PD nanocomposites |
| PCD | particle charge detector |
| NTU | nephelometric turbidity unit |
| TSS | total suspended solids |

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
