# Peer review of "Colorimetric Quantification for Residual Poly-DADMAC in Water Treatment"

_water, doi:10.3390/w15193352_

Round 1
Reviewer 1 Report
I appreciate the authors approach to quantitively analyse the poly-DADMAC (PD) in water. They have tried to present their findings well. But still I have some queries which need to be addressed properly. I recommend a major revision in the manuscript before further consideration by the journal.
1. Write a few words on how the poly-DADMAC (PD) can be leached from membrane? What are the problems related to this leaching?
2. What is the novelty of the work? Why it is important to quantify the concentration of PD in wastewater?
3. A comparison study of the similar works that have been done to quantify PD in wastewater; various strategies that have been implemented; their pros and cons should be clearly mentioned.
4. The author should also mention the possible techniques to remove the PD from wastewater? Is it possible to separate the contaminants by using various types of membrane materials? I advise the authors to go through some recent research articles on membrane-based separation.
[ ACS Appl. Polym. Mater. 9 (2019) 2379-2387; J. Hazard. Mater. 442 (2023) 129955; Sep. Purif. Technol. 279 (2021) 119704-119716; J. Membr. Sci. 522 (2017) 31-44]
5. Is it possible recover PD after complexation with fast green dye? Why the authors have used fast green dye in complexation? What are the other chemicals that have been used in this purpose?
6. The complexation of PD with fast green dye cannot be alone verified with UV analysis. It should be verified with other instrumental techniques like FTIR, EDAX, XPS etc.to
7. The author should add graphical abstract to present their work in a brief manner.
8. Experimental setup should be presented schematically.
9. I have noticed lots of grammatical errors through out the manuscript. Therefore,
I advise the authors to go thoroughly with the manuscript and check the grammatical errors by using standard software.
Must be improved
Reviewer 2 Report
The manuscript entitled “Colorimetric quantification for residual poly-DADMAC in water treatment” is an interesting work on the development of a new and fast method to accurately quantify poly-DADMAC. The work is simple, interesting and well-organized. A few specific comments are below:
1) The complete list of chemicals used for the work is not described in the materials. Example: chemicals for pH control.
2) Please, explain how the detection limit of 3.22 µg L-1 was calculated.
Reviewer 3 Report
In this manuscript, PolyDADMAC (PD) used as effective coagulant in water treatment with colorimetric quantification. Author tested the real waste water samples for the experimental analysis is good.
Author studied the effect of pH , did author check whether temperature affects the results?
Introduction is too brief, include the significance of the work.
Many abbreviations are used without explanation it’s difficult for readers, include the abbreviation section.
Please avoid using words from the title as the keywords- it can narrow the search area/possibility.
There are some grammatical and formatting issues that need further correction.
Authors must need to incorporate future prospective of the presented work in the conclusion part of the manuscript.
Moderate editing of English language required.
Round 2
Reviewer 1 Report
No comment